# Understanding Hierarchical Processes

**DOI:** 10.3390/e24121703

**Published:** 2022-11-22

**Authors:** Wray Buntine

**Affiliations:** 1College of Engineering and Computer Science, VinUniversity, Hanoi 100000, Vietnam; wray.b@vinuni.edu.vn; 2Faculty of Data Science and AI, Monash University, Clayton, VIC 3800, Australia

**Keywords:** Bayesian nonparametrics, Dirichet process, gamma process, Pitman–Yor process, hierarchical process, non-parametric LDA

## Abstract

Hierarchical stochastic processes, such as the hierarchical Dirichlet process, hold an important position as a modelling tool in statistical machine learning, and are even used in deep neural networks. They allow, for instance, networks of probability vectors to be used in general statistical modelling, intrinsically supporting information sharing through the network. This paper presents a general theory of hierarchical stochastic processes and illustrates its use on the gamma process and the generalised gamma process. In general, most of the convenient properties of hierarchical Dirichlet processes extend to the broader family. The main construction for this corresponds to estimating the moments of an infinitely divisible distribution based on its cumulants. Various equivalences and relationships can then be applied to networks of hierarchical processes. Examples given demonstrate the duplication in non-parametric research, and presents plots of the Pitman–Yor distribution.

## 1. Introduction

The hierarchical Pitman–Yor process (HPYP) was first presented as a solution to n-gram language models [1] where it mimics the behavior of the Kneser–Ney algorithm [2]. It is an extension of the hierarchical Dirichlet process (HDP) [3]. The HPYP has since been used in a wide variety of ways, including for previously state-of-the-art and competitive algorithms for topic models [4] and text compression [5]. The HDP has been used for previously state-of-the-art and competitive algorithms for tweet clustering [6] and document segmentation [7]. Many more novel and creative uses of these processes exist, for instance, hierarchical topic models [8]. More general reviews are given by Teh and Jordan [9] and Jordan [10]. The gamma process can also be used hierarchically [11] and provides an alternative scheme for handling the HDP. The notion of hierarchical models fits in well with the computational approach to statistical modelling adopted in the machine learning community.

However, what exactly is the HPYP? A key concept for understanding the HDP and the HPYP is the notion of a discrete base probability measure. The base measure is a source measure for sampling points of the HDP or HPYP. These are discrete just when they have a countable number of possible points (the set on which the measure is based is countable). When finite, the base probability measure is just a probability distribution, usually represented as a vector. However, in non-parametric modelling, we seek to model structured objects for which the dimension may be unknown ahead of time: the number of clusters for points, the depth of a tree, the number of atoms in a molecule, the number of words in a sentence. Allowing the base measure to be countably infinite is a useful abstraction in this situation. Moreover, being able to generate an infinite discrete base probability measure provides us with the ability to model prior distributions for our structured objects without fixing dimensions ahead of time. The above models for text and clustering give examples.

It is known that the hierarchical Dirichlet process, when applied to a finite discrete base distribution, is just a Dirichlet distribution. Indeed, this property is the axiomatic definition of the process [12]. So, applications of and inference with the HDP are really just using hierarchical Dirichlet distributions, requiring no non-parametric theory to describe, although algorithms may be using non-parametric methods.

So, there is a clear concept of what the HDP model is. What is the corresponding result for the hierarchical Pitman–Yor process? For all the algorithms using the HPYP, it would be nice to know what their actual model is! Teh first referred to the hierarchical version of the PYP as the Pitman–Yor distribution [in talks accompanying] [1], saying it has “no known analytical form”. Moreover, is there a more general theory of hierarchical processes, and why does this case (the HDP) come out so neatly? These questions for hierarchical processes have been addressed in recent theory [13,14,15].

Note the Bayesian theory of non-hierarchical processes is extensive. A comprehensive analysis of different processes is developed by James [16], in the more general context of the generalised Indian buffet process [17]. The general posterior analysis of their normalised versions, including the DP, is developed by James et al. [18]. A useful review of theory and a slice sampler for the case of the normalised generalised gamma is given in Lomeli et al. [19]. A study of some of the processes considered here can also be found in Zhou and Carin [11], focusing on gamma processes and their relationships.

However, these treatments are grounded in extensive probability theory and assume the reader is already familiar with Poisson point processes, Lévy processes, subordinators and other advanced areas [20,21]. Some of these details are not strictly necessary for the understanding of the basic ideas. This paper presents the relevant background theory in a self-contained way to develop models for hierarchical processes generally based on the theory of subordinators and completely random measures [20,21]. The theory for the most part reinterprets results from the Bayesian non-parametric and statistical communities [18,22,23], though some related ideas can also be found in machine learning [11]. However, the answers to the questions about the nature of the HPYP and general application to hierarchical processes, networks of hierarchical processes and generalised Chinese restaurants are not well-known outside the Bayesian non-parametric community, so we present them here in a unified manner.

## 2. Background Theory

A formal theory of Poisson point processes (PPP), Lévy processes and completely random measures (CRMs) with treatment of measure theory is needed to rigorously cover this area [20,21]. Here, an informal summary is given, though trying to maintain a degree of precision, for instance keeping adequate rigor in the statement of results.

### 2.1. Completely Random Measures

A CRM is a discrete measure μ(dx) on a space X constructed as
(1)$$\mu \left(x\right)=C_{0}+\sum_{i=1}^{\infty }\lambda_{i}\delta_{x_{i}}\left(x\right)$$
where the $x_{i}\in X$ are called atoms and are assumed distinct, the $\lambda_{i}\in R^{+}$ called jumps, and the background constant $C_{0}$ is zero in our use. This means that $\mu \left(x_{i}\right)=\lambda_{i}$, evaluated at atoms, and $\mu (x_{i})=0$ otherwise. The $\left(\lambda_{i},x_{i}\right)$ are mutually independent random variables, and a finite number of the $x_{i}$ can also be fixed. These conditions ensure the measure is completely random, that is for A,B⊂X, if A∩B=∅ then μ(A)⊥⊥μ(B).

Moreover, suppose the class of CRMs where $C_{0}=0$ in Equation (Equation 1) can be normalised, so $\mu \left(X\right)=\sum_{i=1}^{\infty }\lambda_{i}<\infty$. This yields discrete probability distributions on X represented as μ(x)/μ(X). These are referred to as normalised random measures with independent increments (NRMIs) [18], a concept developed by Kingman [24], and are a general class of discrete probability distributions.

### 2.2. Poisson Point Process

A Poisson point process (PPP) is a stochastic process whose samples represent sets of independent events on a measurable space X. For a sample, the count of events in A⊆X is denoted N(A)∈N. Events are considered to be a countable subset of X, only significant if X is not countable, for instance the real line. The PPP has complete independence, so for A,B⊂X, if A∩B=∅ then N(A)⊥⊥N(B) and N(A∪B)=N(A)+N(B). The sample is specified by a rate ρ(dx) which is any measure on X. In PPP theory, the rate is referred to as a Lévy measure. The PPP has the defining property that N(A)∼Poisson(ρ(A)), and samples can be generated from this by working with an ever finer partition of the space X.

A special class of PPP can be used as a family of priors for a CRM. Assume a PPP has rate ρ(dλ)μ(dx) for $\lambda \in R^{+}$ and x∈X. This is called homogeneous because the terms in λ and *x* are independent [18]. In the case considered here, the μ(dx) is a measure on X called a base measure, and the rate ρ(dλ) has the condition $\int_{0}^{\infty }\min\left(1,\lambda \right)\rho \left(d\lambda \right)<\infty$ to make everything work neatly [20], as follows: This condition is equivalent to $\int_{0}^{\infty }\min\left(\epsilon ,\lambda \right)\rho \left(d\lambda \right)<\infty$ for any 0<ϵ<∞. As a consequence, ρ([ϵ,∞)) is bounded, meaning there will be a finite number of points with λ>ϵ in the sample of the PPP (within a finitely measured subset of X) and $\int_{0}^{\epsilon }\lambda \rho \left(d\lambda \right)$ is bounded, meaning the sum of the λ’s less the ϵ in the sample of the PPP (within a finitely measured subset of X) will be finite even if there is an infinite number of them. Then, a sample from the PPP is a countable set of points which can be used to constuct a CRM.

### 2.3. Example Processes

Consider a number of standard PPPs used to construct CRMs [21]: the generalised (three-parameter) beta process [25], the generalised (three-parameter) gamma process [26] and the stable process. These have the forms given in Table 1, where *M* is a constant background rate. They are given without specifying a base measure on X, which could be given as a final parameter.

The Poisson process and the negative binomial process [11] are also included in Table 1. Both are used in the hierarchical context in Section 4.

The first three processes in Table 1 are widely used in various forms in the non-parametric Bayesian and machine learning communities. From a Bayesian perspective, they are best thought of as improper priors corresponding to the beta, gamma and gamma distributions, respectively. This analysis is presented later in Section 3.4.

NRMIs can be created by normalising CRMs. These are sometimes generated directly from distributions consisting of a normalised discrete set of weights as probabilities. So, generating the $\vec{\lambda }$ according to a generalised (or three parameter) gamma process, GGP(M,α,β), and then normalising yields, what is called a normalised generalised gamma process (NGG). The normalised generalised gamma process (NGG) is constructed analogously to the Dirichlet process, which normalises the gamma process. They represent the main examples of NRMIs. These NRMIs, however, are not paired with base measures when forming a discrete process on X, rather they need to be paired with base distributions Pr(x) since only one point is generated per sample. Denote the NGG process as NGG(α,β,M) or NGG(α,β,M,h(·)), where α,β,M are as described for the GGP, line 3 of Table 1, and h(·) is a base distribution. The DP is effectively the case when α=0.

Traditionally, the parameter vector part of the DP in Equation (Equation 1) is called a GEM distribution (specifically, when a size-biased order is used [27]), named after Griffiths, Engen and McCloskey [28]. This can be represented as an infinite vector $\vec{\lambda }=\left(\lambda_{1},\lambda_{2},\ldots \right)$. Correspondingly, there is a two-parameter version of $\vec{\lambda }$ corresponding to the PYP, GEM(α,β), which has discount 0≤α<1 and concentration β>−α. Then, GEM(0,β) is the original GEM. Including the base distribution h(·) yields DP(β,h(·)) and PYP(α,β,h(·)).

The Pitman–Yor process itself was developed by Pitman and Yor [28], and a general scheme for developing related models is by Pitman [29], called Poisson–Kingman models. However, as to be shown, the hierarchical PYP is very different from the PYP, so this theory is not entirely relevant for the hierarchical case. Alternatively, in Pitman and Yor [28] ([Proposition 21]), it was shown that a PYP can be developed by marginalising out a parameter of the NGG as follows.

**Lemma** **1.**(Deriving a PYP from a NGG) *Let $\mu \left(x\right)∼NGG\left(\alpha ,M,h(\cdot )\right)$ for α,M>0 and suppose M∼gamma(β/α,1) for β>0, then it follows that μ(x)∼PDP(α,β,h(·)).*

The result is presented rather indirectly in Pitman and has been re-expressed by several authors [23] ([Section 3.1.1]), [30] ([Corollary 1]), and leads to a different class of models to the Poisson–Kingman models called Poisson-gamma models [23].

Notice the lemma restricts the PYP to the case where the concentration is positive. More generally, PYPs can have concentration β>−α. When β=0 and α>0, then the PYP is formed from normalising a positive stable distribution.

## 3. Defining Processes Axiomatically

This section gathers together some definitions and theory in order to present a general class of processes built on CRMs that can be treated hierarchically analogous to the Dirichlet process.

### 3.1. Subordinators

A simple useful case of these PPPs has the domain X being $R^{+}$, the positive real line, and is constant for X, so the rate is ρ(dλ) for $\lambda ,x\in R^{+}$. For this, define a new process for our case $C_{0}=0$ given by the cumulative values,
$$\sigma_{t}=\mu \left(\left(0,t\right]\right)=\sum_{i=1}^{\infty }\lambda_{i}\delta_{x_{i}\leq t}$$

So, $\sigma_{0}=0$ and $\sigma_{t}$ increases in steps as each distinct $x_{i}$ is passed. This $\sigma_{t}$ corresponds to the class of so-called pure jump driftless subordinators, which are a kind of nondecreasing Lévy process, which in turn are processes with stationary independent increments [20]. The key relationship that underlies the general theory of these processes is that $\sigma_{t}$ is distributed according to a particular infinitely divisible non-negative distribution, explained in Theorem 1. Examples are given in Table 1. So, for instance, for the generalised gamma process with parameters (M,α,β), the total $\sigma_{1}=\sum_{i=1}^{\infty }\lambda_{i}\delta_{x_{i}\leq 1}$ is distributed as a Tweedie distribution with parameters $\left(\alpha ,M^{1/\alpha },\beta \right)$.

The basic connection is given as follows, a special case of the Lévy–Khintchine formula for subordinators. This uses the Laplace exponent of a 1D random variable *y* defined as the function (of *u*) $\varepsilon [e^{-uy}]$, which is related to the characteristic function.

**Theorem** **1.**
*Consider $\sigma_{t}$ defined as previously by a PPP with rate ρ(dλ) for $\lambda ,x\in R^{+}$ and ρ(dλ) satisfying $\int_{0}^{\infty }\min\left(1,\lambda \right)\rho \left(d\lambda \right)<\infty$. The Laplace exponent of $\sigma_{t}$ is given by*

$$\varepsilon [e^{-u\sigma_{t}}]=e^{-t\psi (u)}$$

*where $\psi \left(u\right)=\int_{\left(0,\infty \right)}\left(1-e^{u\lambda }\right)\rho \left(d\lambda \right)$. This form means that $\sigma_{t}$ has an infinitely divisible non-negative distribution. The t here can be referred to as the parameter for divisibility, occurring in any infinitely divisible distribution.*


Thus, given a rate ρ(dλ) defining a particular $\sigma_{t}$, one can derive its Laplace exponent ψ(u) and then infer the distribution on $\sigma_{t}$ (where analytically possible). Note the scaling term *M* in Table 1 plays the role of *t*.

Some instances of this pairing, an infinitely divisible non-negative distribution with a corresponding rate are given in the last two columns of Table 1. Note that distributions corresponding to the generalised beta process are not well-known. Other distributions that could be included in the table are the inverse beta distribution (the beta distribution is not infinitely divisible but its inverse is), which includes the Pareto and F-distributions, and the generalised inverse gamma distribution [31].

### 3.2. Axiomatic Definitions

To extend Theorem 1 to broader classes of base distributions on general domains X, not just the positive real line with constant measure used in subordinators, one can give an axiomatic definition of a process based on an infinitely divisible non-negative distribution:The derived process is a CRM,The process behaves like the given infinitely divisible distribution on subsets of X.

**Definition** **1.**(Axiomatic definition of a CRM process) *Consider an infinitely divisible non-negative distribution G(μ), where μ is the parameter for divisibility. Further assume its Laplace exponent has zero drift. Given a measurable space X, positive intensity M and measure h(dx) on X, consider a stochastic process denoted GP(M,h(·)) induced by G(μ) as follows. X∼GP(M,h(·)) yields a CRM on X such that*
 *1.* 
*For A,B⊂X, if A∩B=∅ then X(A)⊥⊥X(B),*
 *2.* 
*For A⊆X, X(A)∼G(Mh(A)).*



The first condition implies that the measures are CRMs as per Equation (Equation 1). The second condition implies one can construct the discrete measures iteratively, on an ever finer, nested sequence of partitions using the distribution G(). Alternatively, one can use the Lévy–Khintchine formula of Theorem 1 to show the existence of a corresponding rate yielding a CRM with rate Mh(dx)ρ(dλ) which must then satisfy the conditions.

Note that the Dirichlet process can be defined axiomatically [12], akin to Definition 1 with the Dirichlet distribution used instead of the gamma distribution, and base probability distribution used instead of a base measure. This axiomatic construction generalises for any infinitely divisible non-negative distribution as follows:

**Definition** **2.**(Axiomatic definition of an NRMI process) *Consider an infinitely divisible non-negative distribution G(μ), where μ is the parameter for divisibility. Further assume its Laplace exponent has zero drift. Consider as well the distribution on probability vectors induced by generating K values $\zeta_{k}∼G\left(\mu_{k}\right)$ and normalising to obtain*
$$\left(\frac{\zeta_{1}}{\sum_{k=1}^{K}\zeta_{k}},\cdots ,\frac{\zeta_{K}}{\sum_{k=1}^{K}\zeta_{k}}\right)\;.$$
*Denote this distribution by ${NG}_{K}\left(\vec{\mu }\right)$, where $\vec{\mu }$ is the vector of K value $\mu_{k}$, given a measurable space X, positive intensity M and probability distribution h(dx) on X. A process denoted NGP(M,h(·)), developed from G(μ), is defined as follows. It is a stochastic process whose sample is a probability measure on X such that if C∼NGP(M,h(·)) then for any finite partition $A_{1},\ldots ,A_{K}$ of X, and count N>0, $\left(C\left(A_{1}\right),\ldots ,C\left(A_{K}\right)\right)∼multinomial\left(N,{NG}_{K}\left(Mh\left(A_{1}\right),\ldots ,Mh\left(A_{K}\right)\right)\right)$.*


In this way, a multinomial process can be defined axiomatically, as done by Zhou and Carin [11] [Corollary IV2]. One uses MP(N,h(·)) where $N\in N^{+}$ is the total count and h(·) a probability measure. The axiomatic part is $\left(X\left(A_{1}\right),\ldots ,X\left(A_{K}\right)\right)∼\mathrm{multinomial}(N,$$\left(h\left(A_{1}\right),\ldots ,h\left(A_{K}\right))\right)$. Similarly, a Dirichlet compound multinomial (DCM) process can be defined, denoted as DCMP(N,h(·)), where the axiomatic part is $\left(X\left(A_{1}\right),\ldots ,X\left(A_{K}\right)\right)∼\mathrm{DCM}\left(N,\left(h\left(A_{1}\right),\ldots ,h\left(A_{K}\right)\right)\right)$. These correspond to a PPP and a NBP, respectively, both given in Table 1, where one has also conditioned on the total count being *N*.

### 3.3. On the Tweedie Distribution

From Table 1, the marginal distribution for the generalised gamma process is the Tweedie distribution [32] with exponent α, or sometimes expressed as index $p=1+\frac{1}{1-\alpha }$ which has p>2 necessarily. For α=0, the Tweedie distribution becomes a gamma distribution.

The Tweedie distribution with exponent 0<α<1 is formed from a positive stable distribution defined in terms of the stable distribution with characteristic exponent α, scale parameter $s=M^{1/\alpha }$ location zero and symmetry one [33]. This distribution, denoted as pstable(α,s), has the functional form [adding a scale to the standard formula of] [34] given by the remarkable formula
$$\begin{array}{ccc} \mathrm{Pr}(x|pstable(\alpha ,s)) & = & \frac{\alpha }{1-\alpha }\frac{1}{s\pi }{\left(x/s\right)}^{-\frac{1}{1-\alpha }}\int_{0}^{\pi }a_{\alpha }\left(\nu \right)e^{-{\left(x/s\right)}^{-\frac{\alpha }{1-\alpha }}a_{\alpha }\left(\nu \right)}d\;\nu \\ where\;a_{\alpha }\left(\nu \right) & = & \frac{\sin\left(\left(1-\alpha \right)\nu \right){\left(\sin\left(\alpha \nu \right)\right)}^{\alpha /(1-\alpha )}}{{\sin\left(\nu \right))}^{1/(1-\alpha )}}\;, \end{array}$$
which yields a simple ingenious sampling formula [34]. To obtain a Tweedie distribution, “exponentially tilt” the pstable(α,s), calculated by multiplying by $e^{-\beta x}$ and renormalising. The construction of exponentially tilting the distribution (see for instance Pitman [29]) gives the following: $$\mathrm{Pr}(x|\mathrm{Twe}(\alpha ,s,\beta ))=e^{{\left(s\beta \right)}^{\alpha }-\beta x}\mathrm{Pr}(x|pstable(\alpha ,s))\;.$$

Here, the term $e^{{\left(s\beta \right)}^{\alpha }-\beta x}$ is added to achieve normalisation.

### 3.4. Bayesian Analysis

A complete Bayesian analysis of CRMs and NRMIs has been developed by James [16] and James et al. [18], respectively, in the non-hierarchical context. This models the standard framework in which hierarchical DPs or hierarchical PYPs are used, but also applies to the Indian buffet process [17]. This is informally developed below so that their theoretical results can be subsequently used. By Bayesian analysis, the following is meant: one has an infinitely divisible distribution suitable for use with Theorem 1. One samples a CRM from this with unknown parameters of rates $\vec{\lambda }$ and atoms $x_{i}$. Now, hierarchically sample sets of atoms from this CRM using a PPP. Each hierarchical sample from the CRM is a discrete set A⊆X, and multiple samples are drawn. Then, the task is to estimate the parameters of the parent CRM.

A CRM is represented in the form $\mu \left(x\right)=\sum_{i=1}^{\infty }\lambda_{i}\delta_{x_{i}}\left(x\right)$ for x∈X where the $x_{i}$ are distinct and is generated according to a homogeneous PPP with rate ρ(dλ)ω(dx) where ρ(dλ) is a rate satisfying the conditions of Theorem 1. One then takes *J* samples from this according to a PPP, so ${\vec{n}}_{j}∼\mathrm{PPP}\left(\mu \left(\cdot \right)\right)$ for j=1,…,J. Each sample will be a finite subset of the atoms, some possibly occurring multiple times. For representational purposes, post hoc reorder the atoms of μ(x) so that only the first *I* have non-zero counts. So, for I<i≤∞, none of the samples ${\vec{n}}_{j}$ contain $x_{i}$. The count of atom $x_{i}$ in sample *j* is represented as $n_{j,i}$, so the condition ${\vec{n}}_{1:J,i}\neq \vec{0}$ means that at least one of the *J* samples contains an atom $x_{i}$.

The following informal analysis is offered as an explanation, but formal proofs are in James [16]. To make analysis feasible, we have to convert the rate ρ(dλ) to one with finite total measure. James [16] ingeniously and elegantly presents this by viewing the posterior for $\mu_{i}$ after seeing the evidence of having at least one non-zero value in the *J* values, so ${\vec{n}}_{1:J,i}\neq \vec{0}$. For the particular sampling distribution of $n_{j,i}$, in our case a $\mathrm{Poisson}(\lambda_{i})$,
$$\mathrm{Pr}({\vec{n}}_{1:J,i}\neq \vec{0}|\lambda_{i})=1-e^{-J\lambda_{i}}$$
which has a term in $\lambda_{i}$ so the posterior rate $\mathrm{Pr}({\vec{n}}_{1:J,i}\neq \vec{0}|\lambda_{i})\rho \left(d\lambda_{i}\right)$ obtains finite total measure. Denote this total by $\Psi_{J}=\int \mathrm{Pr}({\vec{n}}_{1:J,i}\neq \vec{0}|\lambda_{i})\rho \left(d\lambda_{i}\right)$. Then, working entirely with finite PPPs, one can compute the marginal. First, we generate the number of non-zero atoms *I* (for the given sample count *J*) by a Poisson and then generate the vector of counts for each atom ${\vec{n}}_{1:J,i}$, like so
(2)$$\begin{array}{ccc} \mathrm{Pr}({\vec{n}}_{1},\ldots ,{\vec{n}}_{J}|\rho \left(d\lambda \right),\mathrm{PPP}) & = & e^{-\Psi_{J}}\frac{\Psi_{J}^{I}}{I!}\prod_{i=1}^{I}\mathrm{Pr}({\vec{n}}_{1:J,i}|{\vec{n}}_{1:J,i}\neq \vec{0})\\ & = & e^{-\Psi_{J}}\frac{\Psi_{J}^{I}}{I!}\prod_{i=1}^{I}\frac{\int \mathrm{Pr}({\vec{n}}_{1:J,i}|\lambda )\rho \left(d\lambda \right)}{\int \mathrm{Pr}({\vec{n}}_{1:J,i}\neq \vec{0}|\lambda )\rho \left(d\lambda \right)}\\ & = & e^{-\Psi_{J}}\frac{1}{I!}\prod_{i=1}^{I}\int \mathrm{Pr}({\vec{n}}_{1:J,i}|\lambda )\rho \left(d\lambda \right)\;, \end{array}$$
where the term I! can be removed if one considers that the atoms are ordered. With similar reasoning, one obtains:**the posterior rate of $\lambda_{i}$:** for i≤I has rate $\mathrm{Pr}({\vec{n}}_{1:J,i}|\lambda_{i})\rho \left(d\lambda_{i}\right)$,**the posterior rate of the remainder CRM:**$\mu_{R}\left(x\right)=\sum_{i=I+1}^{\infty }\lambda_{i}\delta_{x_{i}}\left(x\right)$, has rate$\mathrm{Pr}({\vec{n}}_{1:J}\neq \vec{0}|\lambda )\rho \left(d\lambda \right)\omega \left(dx\right)$,**the total rate of the remainder CRM:**$T_{R}=\sum_{i=I+1}^{\infty }\lambda_{i}$ as given by Theorem 1.

The key formula for this kind of analysis is given in our context in Table 2.

The first line, the beP-BP case, is the three parameter beta process with Bernoulli data, which is the three parameter Indian buffet process. The second line is the gamma process with Poisson data. Note the data marginals $\int \mathrm{Pr}({\vec{n}}_{1:J,i}|\lambda )\rho \left(d\lambda \right)$ in our context can be obtained more directly, developed in Section 4.2, so formulas are not given.

## 4. Using Discrete Base Distributions

It is important to understand what happens when you use a discrete distribution as a base distribution to a CRM, since this is what happens when hierarchical constructions of these processes are made. Let the base measure on X have the form $\mu \left(x\right)=\sum_{i=1}^{\infty }\lambda_{i}\delta_{x_{i}}\left(x\right)$, and the CRM is constructed using a homogeneous PPP with rate ρ(dλ)ω(dx). What happens? This section considers various implications of this. Note different but more extensive treatment of this scenario for the results on moments, Section 4.2, and the generalised Chinese restaurant process, Section 4.4, is given by Camerlenghi et al. [14], Argiento et al. [15]. They also include example MCMC sampling algorithms.

### 4.1. General Results

Superposition of PPPs says to decompose a discrete CRM into a union of trivial PPPs each with rate in the form $\mu_{i}\rho \left(\lambda \right)\delta_{x_{i}}$, so the X component is a delta function. The resultant CRM is also trivial and takes the form, using Definition 1, $\Lambda \delta_{x_{i}}$, where Λ is the total of the $\lambda_{k}$ generated using the rate $\mu_{i}\rho \left(\lambda \right)$. This total is distributed as the corresponding marginal distribution for the subordinator with intensity parameter $\mu_{i}$, as per Theorem 1.

**Lemma** **2.**(CRM when base measure is discrete) *Let a discrete measure on X have the form $\mu \left(x\right)=\sum_{i=1}^{\infty }\mu_{i}\delta_{x_{i}}\left(x\right)$ for x∈X where the $x_{i}$ are distinct, and a homogeneous CRM is constructed by sampling using a PPP with rate ρ(dλ)μ(dx) on $R^{+}\times X$. Let Γ(t) be the marginal total distribution for the corresponding subordinator, where t is the parameter of divisability. Then, the CRM has the form*
(3)$$\gamma \left(x\right)=\sum_{i=1}^{\infty }\gamma_{i}\delta_{x_{i}}\left(x\right)$$
*where the random variable $\gamma_{i}∼\Gamma \left(\mu_{i}\right)$, and the $x_{i}$ are inherited from μ(·).*

The CRM μ(·) when used as a base distribution for a PPP is mapped element-wise to form a new CRM γ(·). So, no PPP modelling is required if you know the form of the element-wise distribution.

There are a number of very convenient and well-known properties of the Dirichlet that allow it to be used in hierarchical contexts. As it happens, most of these properties also hold for other NRMIs with discrete base measures, and some for CRMs, so these results are developed here. The first property is aggregation. This has that if $\left(x_{1},x_{2},x_{3}\right)∼\mathrm{Dirichlet}\left(\alpha_{1},\alpha_{2},\alpha_{3}\right)$, then $\left(x_{1},x_{2}+x_{3}\right)∼\mathrm{Dirichlet}\left(\alpha_{1},\alpha_{2}+\alpha_{3}\right)$, and this applies for a Dirichlet of any dimension. The second property is renormalisation and has that if $\left(x_{1},x_{2},x_{3}\right)∼\mathrm{Dirichlet}\left(\alpha_{1},\alpha_{2},\alpha_{3}\right)$ then $\left(x_{1},x_{2}\right)/\left(x_{1}+x_{2}\right)∼\mathrm{Dirichlet}\left(\alpha_{1},\alpha_{2}\right)$. Both properties clearly follow from the fact that a Dirichlet is a normalised Gamma, and by analogy hold for NRMIs too.

**Definition** **3.**(Aggregation property) *Consider a process that takes a measure as an input parameter and outputs another measure. The process has the aggregation property if when $\sum_{i=1}^{\infty }\gamma_{i}\delta_{x_{i}}\left(x\right)$ is a sample from the process with a discrete input measure $\sum_{i=1}^{\infty }\mu_{i}\delta_{x_{i}}\left(x\right)$ where the $x_{i}$ are distinct, then $\sum_{i=3}^{\infty }\gamma_{i}\delta_{x_{i}}\left(x\right)+\left(\gamma_{1}+\gamma_{2}\right)\delta_{x_{1}}\left(x\right)$ is a sample from the process with input measure $\sum_{i=3}^{\infty }\mu_{i}\delta_{x_{i}}\left(x\right)+\left(\mu_{1}+\mu_{2}\right)\delta_{x_{1}}\left(x\right)$.*

The aggregation property can be used to form arbitrary groupings of the dimensions.

**Definition** **4.**(Renormalisation property) *Consider a process that takes a measure as an input parameter and outputs a probability measure. The process has the renormalisation property if when $\sum_{i=1}^{\infty }\gamma_{i}\delta_{x_{i}}\left(x\right)$ is a sample from the process with a discrete input measure $\sum_{i=1}^{\infty }\mu_{i}\delta_{x_{i}}\left(x\right)$ where the $x_{i}$ are distinct, then $\frac{1}{\sum_{i=2}^{\infty }\gamma_{i}}\sum_{i=2}^{\infty }\gamma_{i}\delta_{x_{i}}\left(x\right)$ is a sample from the process with input measure $\sum_{i=2}^{\infty }\mu_{i}\delta_{x_{i}}\left(x\right)$.*

The renormalisation property then yields probability measures on subsets of the discrete domain, so it can be used for incremental sampling.

**Lemma** **3.**(Aggregation and renormalisation) *Consider the context of Lemma 2. The aggregation property holds for all CRMs and NRMIs. In the case of an NRMI, the renormalisation property holds. For the PYP, the aggregation property holds but not the renormalisation property.*

The results for the PYP can be developed using Lemma 1. The aggregation and renormalisation properties together mean that efficient size-biased samplers can be developed for NRMIs by sampling one dimension at a time according to a two-dimensional version of the NRMI, which is effectively the stick breaking construction (although, only a few explicit cases of this are known). Alternatively, one can sample the underlying CRM according to its corresponding infinitely divisible distribution.

A third property of the Dirichlet is neutrality, which applies in the context of renormalisation and requires that the part taken away is independent of the remainder: if $\left(x_{1},x_{2},x_{3}\right)∼\mathrm{Dirichlet}\left(\alpha_{1},\alpha_{2},\alpha_{3}\right)$, then $\left(x_{1},x_{2}\right)/\left(x_{1}+x_{2}\right)$ is independent of $x_{3}$.

**Definition** **5.**(Neutrality property) *Consider a process that outputs a finite discrete probability measure, and without loss of generality let $\sum_{i=1}^{I}\gamma_{i}\delta_{x_{i}}\left(x\right)$ be a sample from the process where the $x_{i}$ are distinct. The process is completely neutral if there exists mutually independent non-negative variables $\lambda_{1},\ldots ,\lambda_{I}$ such that $\left(\gamma_{1},\ldots ,\gamma_{K}\right)$ and $\left(\lambda_{1},\lambda_{2}\left(1-\lambda_{1}\right),\ldots ,\lambda_{I}\prod_{i=1}^{I-1}\left(1-\lambda_{i}\right)\right)$ have the same distribution.*

It is known that the only distribution on finite probability vectors with complete neutrality is the Dirichlet distribution [35].

### 4.2. Results on Moments

Moments of CRMs are critical quantities for their posterior analysis [18,36] to be developed in Section 5 and seen in Section 3.4. The generalised version is derived by unfolding the recursion that relates the moments of a distribution to its cumulants. In the context of Lemma 2, where $\gamma_{i}∼\Gamma \left(\mu_{i}\right)$, various moments such as $\varepsilon \left[\gamma_{i}^{n}|\mu_{i}\right]$ and $\varepsilon \left[\gamma_{i}^{n}e^{-U\gamma_{i}}|\mu_{i}\right]$ can be computed recursively from the moments of the PPP rate ρ(dλ) [22] ([Section 1.3]) and its exponentially titled form. Note these moments compute the marginals one needs for multinomial and Poisson data, respectively, hence their importance.

In the theorem, the notation $P^{n}$ is used to represent all possible non-empty partitions of *n* items, the set {1,…,n}. As an example, $P^{3}$ is the set
{{{1,2,3}},{{1},{2,3}},{{1,2},{3}},{{1,3},{2}}},{{1},{2},{3}}},
so it contains the partition {{1},{2,3}} as an element, for instance. Moreover, $P_{K}^{n}\subseteq P^{n}$ are all members are of size *K*, so $|P_{1}^{n}\left|=\right|P_{n}^{n}|=1$ and $|P_{2}^{3}|=3$.

The following Lemma is a corollary the major result by Pitman [22], and some related results appear in Camerlenghi et al. [14], as proven in Appendix A.

**Lemma** **4.**(CRM moments when base measure is discrete) *Consider the context of Lemma 2. Let $\kappa_{n}=\int_{0}^{\infty }\lambda^{n}\rho \left(d\;\lambda \right)$ be the n-th moment for rate ρ(λ), where it exists for $n\in N^{+}$. Let ψ(t) be the Laplace exponent for the rate. Then, the n-th cumulant of $\gamma_{i}$ can be re-expressed as a moment of the original rate ρ(λ), and the n-th moment of $\gamma_{i}$ is computed recursively from it.*
(4)$$\begin{array}{ccc} \kappa_{n} & = & {\left(-1\right)}^{n+1}{\left.\frac{d{\;}^{n}\psi \left(t\right)}{d\;t^{n}}\right|}_{t=0} \end{array}$$
(5)$$\begin{array}{ccc} {cumulant}_{n}\left(\gamma_{i}\right) & = & \mu_{i}\kappa_{n} \end{array}$$
(6)$$\begin{array}{ccc} \varepsilon \left[\gamma_{i}^{n}|\mu_{i}\right] & = & \sum_{\Pi \in P^{n}}\mu_{i}^{\left|\Pi \right|}\prod_{C\in \Pi }\kappa_{\left|C\right|} \end{array}$$
(7)$$\begin{array}{ccc} & = & \sum_{K=1}^{n}\mu_{i}^{K}\;T_{K}^{n} \end{array}$$
(8)where TKn=∑Π∈PKn∏C∈Πκ|C|=∑k=1n−K+1TK−1n−kn−1k−1κk.

Note the recursion for $T_{K}^{n}$ starts at $T_{1}^{n}=\kappa_{n}$ derived from the non-recursive form.

Thus, if the Laplace exponent is known, one can usually compute the moments of the process and hence the cumulants and evidence terms for its corresponding infinitely divisible distribution. When one has Poisson data, required moments need to include an exponential term, as proven in Appendix B.

**Corollary** **1.**(Adding an exponential term) *Consider the context of Lemma 4 with rate ρ(λ). To obtain exponentiated moments of the form $\varepsilon \left[\gamma_{i}^{n}e^{-U\gamma_{i}}|\mu_{i}\right]$, complete the following steps.*
 *1.* 
*Use rate $e^{-U\lambda }\rho \left(\lambda \right)$, and the Laplace exponent is given by ψ(U+t)−ψ(U), so the corresponding moments are given by*

$$\kappa_{n,U}={\left(-1\right)}^{n+1}{\left.\frac{d{\;}^{n}\psi \left(t\right)}{d\;t^{n}}\right|}_{t=U}$$

 *2.* 
*Obtain the corresponding $T_{K}^{n}$ using Equation (Equation 8) with the $\kappa_{n,U}$, denoted $T_{K,U}^{n}$.*
 *3.* 
*Consequently,*

$$\varepsilon [\gamma_{i}^{n}e^{-U\gamma_{i}}|\mu_{i}]=e^{-\mu_{i}\psi \left(U\right)}\sum_{K=1}^{n}\mu_{i}^{K}\;T_{K,U}^{n}\;.$$




The components from Lemma 4 for the processes in Table 1 are given in Table 3. These appear in various places in the broader statistical literature. The Laplace exponent is usually computed using integration by parts. The form $S_{s,\alpha }^{n}$ is the second order generalised Stirling number used in PYP inference [1,37], a generalized Stirling number of type (−1,−d,0) [38]. It can be verified using its recursion [37] with Equation (Equation 8).

Note the general beta process has no simple analytic form for either ψ(t) or its marginal distribution. Fortunately, is is difficult to envisage a situation where it would be used hierarchically.

### 4.3. The Gamma Process

Let us consider the simple example of a gamma process, GP(M,β) and assume data yields Poisson likelihoods in the form $\prod_{i=1}^{I}\gamma_{i}^{n_{i}}e^{-U\gamma_{i}}$ for dimensions i=1,…,I in the context of Lemma 2. The marginal likelihood then, for the $\mathrm{data}=\{\;n_{i},x_{i}:\;i=1,\ldots ,I\;\}$ is given by
$$\mathrm{Pr}(data|\mu (\cdot ))=\varepsilon [e^{-U\gamma_{R}}|\mu_{R}]\prod_{i=1}^{I}\varepsilon [\gamma_{i}^{n_{i}}e^{-U\gamma_{i}}|\mu_{i}]$$
where the expectation is taken with respect to γ(·)∼GP(M,β,μ(·)), which has $\gamma_{i}∼\mathrm{gamma}\left(M\mu_{i},\beta \right)$. Note, in this case, the exact solution is known since the data marginals of the gamma distribution have a simple closed form,
(9)$$\varepsilon [\gamma_{i}^{n_{i}}e^{-U\gamma_{i}}|\mu_{i}]=\int_{0}^{\infty }\gamma^{n_{i}}e^{-U\gamma }\frac{\beta^{M\mu_{i}}}{\Gamma (M\mu_{i})}\gamma^{M\mu_{i}-1}e^{-\beta \gamma }d\;\gamma =\frac{\Gamma (M\mu_{i}+n_{i})}{{\left(U+\beta \right)}^{M\mu_{i}+n_{i}}}\frac{\beta^{M\mu_{i}}}{\Gamma (M\mu_{i})}$$

Consider, however, using Corollary 1. In this case, moments including $e^{-U\gamma_{i}}$ are found to be $\kappa_{n}=\int_{0}^{\infty }\gamma^{n}e^{-U\gamma }\rho \left(d\;\gamma \right)=M\frac{\Gamma (n)}{{\left(U+\beta \right)}^{n}}$, and the Laplace exponent can be obtained using integration by parts as Mlog(1+t/β). One can confirm that the corresponding index $T_{K}^{n}=\frac{1}{{\left(U+\beta \right)}^{n}}S_{K}^{n}M^{K}$ where $S_{K}^{n}$ is an unsigned Stirling number of the first kind, an index that is found in collapsed versions of the CRP. Equation (Equation 8) yields the standard recurrence for it. So, by Equation (Equation 7), and adding back the term $e^{-\mu_{i}\psi \left(U\right)}={\left(\frac{\beta }{U+\beta }\right)}^{M\mu_{i}}$ as per Corollary 1, obtain for atom index *i* the moment
(10)$$\mathrm{Pr}(\gamma_{i}^{n}e^{-U\gamma_{i}}|\mu_{i})=\varepsilon [\gamma_{i}^{n}e^{-U\gamma_{i}}|\mu_{i}]=\frac{\beta^{M\mu_{i}}}{{\left(U+\beta \right)}^{M\mu_{i}+n_{i}}}\sum_{K=1}^{n_{i}}S_{K}^{n_{i}}{\left(M\mu_{i}\right)}^{K}\;.$$

The sum can be converted using a standard identity [37] ([Lemma 16]) to get back The sum in Equation (Equation 10) has an interpretation as a form of Chinese restaurant process for the dimension *i*. Each partition of the set $\left\{1,\ldots ,n_{i}\right\}$, given by $\Pi_{i}\in P^{n_{i}}$ corresponds to a configuration of the $n_{i}$ data in $|\Pi_{i}|$ tables. For any table with participants $C\in \Pi_{i}$, the probability of the table is $M\mu_{i}\frac{\Gamma (|C|)}{{\left(U+\beta \right)}^{\left|C\right|}}$. The probability of this configuration $\Pi_{i}$ is then $\prod_{C\in \Pi_{i}}M\mu_{i}\frac{\Gamma (|C|)}{{\left(U+\beta \right)}^{\left|C\right|}}$. So, introducing the partition $\Pi_{i}$ or its size as an additional variable,
$$\begin{array}{ccc} \mathrm{Pr}(\gamma_{i}^{n}e^{-U\gamma_{i}},\Pi_{i}|\mu_{i}) & = & \frac{\beta^{M\mu_{i}}}{{\left(U+\beta \right)}^{M\mu_{i}+n_{i}}}{\left(M\mu_{i}\right)}^{|\Pi_{i}|}\prod_{C\in \Pi_{i}}\Gamma \left(|C|\right)\\ \mathrm{Pr}(\gamma_{i}^{n}e^{-U\gamma_{i}},|\Pi_{i}\left|=K\right|\mu_{i}) & = & \frac{\beta^{M\mu_{i}}}{{\left(U+\beta \right)}^{M\mu_{i}+n_{i}}}S_{K}^{n_{i}}{\left(M\mu_{i}\right)}^{K}\;. \end{array}$$

The second form, the probability of all configurations of size $K(=|\Pi_{i}|)$, follows from Equation (Equation 8).

### 4.4. General Chinese Restaurant Processes

Motivated by the gamma process example just given, now construct a generalised CRP interpretation of the results in Section 4.2. The marginals have an interpretation as generalised versions of Chinese restaurants, including the more efficient collapsed versions [6], both developed in this section. This is intended to complement the comprehensive Bayesian analysis already developed for the non-hierarchical cases by [16,18].

The significance of the formula in Lemma 4 is that the sum in Equation (Equation 6) is over partitions Π of the *n* data points, and $\kappa_{\left|C\right|}$ represents the probability of generating a single element *C* of size |C| (in the partition Π) according to the rate ρ(λ). The sum in Equation (Equation 7) is now over partition sizes *K*, and $T_{K}^{n}$ is the probability of generating a partition of *K* non-empty sets according to the rate ρ(λ).

**Lemma** **5.**(General Chinese restaurant processes for CRMs) *Consider the posterior data marginal for γ(·), as in Corollary 2, where data is in the form of a Poisson likelihood with counts $n_{i}>0$ at each atom $x_{i}$:*
$$\mathrm{Pr}(\left\{n_{i},x_{i}\;:\;i=1,\ldots ,I\right\}|\gamma \left(\cdot \right),U)=\prod_{i=1}^{I}\gamma_{i}^{n_{i}}e^{-U\gamma_{i}}$$
*One can treat $\Pi_{i}\in P^{n_{i}}$ as a latent variable, which represents the seating configuration for instances of the atom. Then, the data marginal using $\Pi_{1},\ldots ,\Pi_{I}$ takes the form:*

(11)
$$\mathrm{Pr}(\left\{n_{i},x_{i},\Pi_{i}\;:\;i=1,\ldots ,I\right\}|\mu \left(\cdot \right),U(=e^{-\psi \left(U\right)\sum_{i=1}^{\infty }\mu_{i}}\prod_{i=1}^{I}\left(\mu_{i}^{|\Pi_{i}|}\prod_{C\in \Pi_{i}}\kappa_{|C_{i}|,U}\right)\;.$$


*Moreover, for any j (including j>I),*

(12)
$$\mathrm{Pr}(x_{j}|\left\{n_{i},x_{i},\Pi_{i}\;:\;i=1,\ldots ,I\right\},\mu \left(\cdot \right))=\mu_{j}\kappa_{1,U}+\sum_{C\in \Pi_{j}}\frac{\kappa_{|C|+1,U}}{\kappa_{|C|,U}}\;,$$

*where the convention is used that $\Pi_{j}=\emptyset$ for j>I (when there is no data). Alternatively, if $K_{i}$, the number of tables for atom index i is handled as a latent variable, then the data marginal given table numbers takes the form:*

(13)
$$\mathrm{Pr}(\left\{n_{i},x_{i},K_{i}\;:\;i=1,\ldots ,I\right\}|\mu \left(\cdot \right),U)=e^{-\psi \left(U\right)\sum_{i=1}^{\infty }\mu_{i}}\prod_{i=1}^{I}\mu_{i}^{K_{i}}\;T_{K_{i},U}^{n_{i}}\;.$$



Equation (Equation 12) is related to the generalized Blackwell–MacQueen sampling scheme by James et al. [18] [Section 3.3]. The data marginals in Equations (Equation 11) and (Equation 13) have a simple Poisson likelihood in $\vec{\mu }$. Thus, a CRP interpretation of a Gamma process can be used for hierarchical inference with a Gamma distribution, as used by Zhou and Carin [11], for instance.

To develop a corresponding formula for NRMIs where they are generated by normalising a CRM, we use an ingenious technique for normalising a CRM within a posterior analysis from [18] The basic idea is to convert multinomial sampling into Poisson sampling (without normalisation) but require some post manipulation to derive the results. A generative variation of this goes as follows:For each multinomial $\vec{n}$ according to the unnormalised values $\vec{\lambda }$, introduce a scale-free latent relative mass denoted *U*, with the scale-invariant improper prior $\frac{d\;U}{U}$.Generate the data needed according to Poisson $n_{i}∼\mathrm{Poisson}\left(U\lambda_{i}\right)$ for i=1,…,∞, noting that $n_{i}=0$ for i>I.Then, the joint posterior on $\vec{n},\vec{\lambda },U$ becomes quite concentrated for *U* and can be marginalised out.To correct the formulas, multiply the marginal by $N=\sum_{i=1}^{I}n_{i}$ to obtain a conversion to a multinomial.

To see that this indeed does what is required, one needs to verify the following identity.
N∫R+∏i=1∞e−Uλi(Uλi)nini!dUU=Nn→∏i=1Iλi∑iλini.

Note the product $\prod_{i=1}^{\infty }$ is well-defined because $\sum_{i=1}^{\infty }\lambda_{i}$ is finite.

**Corollary** **2.**(General Chinese restaurant processes for NRMIs) *Consider the posterior data marginal for γ(·) as given in Lemma 2, where data is in the form of a multinomial likelihood with counts $n_{i}>0$ at each atom $x_{i}$:*
$$\mathrm{Pr}(\left\{n_{i},x_{i}\;:\;i=1,\ldots ,I\right\}|\gamma \left(\cdot \right))=\prod_{i=1}^{I}{\left(\frac{\gamma_{i}}{\sum_{i=1}^{\infty }\gamma_{i}}\right)}^{n_{i}}\;,$$
*and let $N=\sum_{i=1}^{I}n_{i}$ be the total count. Let $U∼gamma(N,\sum_{i=1}^{\infty }\gamma_{i})$. Then, the data marginal using $\Pi_{1},\ldots ,\Pi_{I}$, similarly to Lemma 5, takes the form:*
(14)$$\mathrm{Pr}(\left\{n_{i},x_{i},\Pi_{i}\;:\;i=1,\ldots ,I\right\},U|\mu \left(\cdot \right))=\frac{U^{N-1}}{\Gamma (N)}e^{-\psi \left(U\right)\sum_{i=1}^{\infty }\mu_{i}}\prod_{i=1}^{I}\left(\mu_{i}^{|\Pi_{i}|}\prod_{C\in \Pi_{i}}\kappa_{|C_{i}|,U}\right)\;.$$
*Moreover, for any j (including j>I),*

(15)
$$\mathrm{Pr}(x_{j}|\left\{n_{i},x_{i},\Pi_{i}\;:\;i=1,\ldots ,I\right\},U,\mu \left(\cdot \right))=\mu_{j}\kappa_{1,U}+\sum_{C\in \Pi_{j}}\frac{\kappa_{|C|+1,U}}{\kappa_{|C|,U}}\;.$$


*Alternatively, if each $K_{i}$ is handled as a latent variable, then the data marginal given table numbers takes the form:*

(16)
$$\mathrm{Pr}(\left\{n_{i},x_{i},K_{i}\;:\;i=1,\ldots ,I\right\},U|\mu \left(\cdot \right))=\frac{U^{N-1}}{\Gamma (N)}e^{-\psi \left(U\right)\sum_{i=1}^{\infty }\mu_{i}}\prod_{i=1}^{I}\mu_{i}^{K_{i}}\;T_{K_{i},U}^{n_{i}}\;.$$



Note, to complete the analysis, one needs to model the unseen parts of the processes. So, while it is assumed $\mu_{i}$ for i=1,…,I is being sampled or estimated, of $\mu_{i}$ and $\gamma_{i}$ for i=I+1,…,∞ only a finite number, if any, can be sampled or estimated. Handling these is illustrated in Section 5 using a remainder term $\mu_{R}=\sum_{j=I+1}^{\infty }\mu_{j}$.

In general, then, there are two different levels of inference one can use when the marginal does not have a simple closed form and must instead be computed using the latent forms in Lemma 5 or Corollary 2:

#### Sampling over table configurations:

For the DP, this is exhibited by the standard CRP. One can see from Equations (Equation 6) and (Equation 12) that to resample which table a point belongs to, one would use the following proportionalities: (17)$$\mathrm{Pr}(C|\Pi ,\mu_{k},\ldots )\propto \left\{\begin{array}{cc} \mu_{k}\kappa_{1} & start\;a\;new\;table\\ \frac{\kappa_{|C|+1}}{\kappa_{\left|C\right|}} & add\;to\;table\;C \end{array}\right.\;.$$

#### Sampling over table sizes:

For the PYP, this is demonstrated by table indicator sampling methods [6,39] and “direct” Gibbs sampling of Gasthaus and Teh [5], though subsequently not used because in their context they needed to constantly resample discount α. This is a collapsed sampler that instead samples *K*, the number of tables using Equation (Equation 7): (18)$$\mathrm{Pr}(K|\mu_{k},\ldots )\propto \mu_{k}^{K}T_{K}^{n}$$

This is only efficient when $T_{K}^{n}$ can be tabulated. In the general case, this requires $O(n^{2}K)$ steps to follow using Equation (Equation 8) and O(nK) for cases such as the gamma process above where a simpler double recursion is available for $T_{K}^{n}$ since they are generalised second-order Stirling numbers.

## 5. Variants of the Generalised Gamma Process

In this section, we develop both the CRM and NRMI variants of the generalised gamma process in the hierarchical context. Using the generalised gamma process in an NRMI yields an NGG or a PYP. When the NGG process and the PYP are supplied discrete base distributions as input, they behave analogously to the Dirichlet distribution, as illustrated with Lemma 3. In this discrete context, refer to the corresponding distributions as the NGG distribution and the Pitman–Yor distribution (PYD). Here analytical forms of the PY distribution are developed.

### The Hierarchical Context

Consider an NRMI in the context of the base distribution μ(x), as before. Suppose multinomial type data is observed in the form of counts $n_{k}$ associated with the atoms $x_{k}$ of μ(·) for k=1,…,K, with total count $N=\sum_{k=1}^{K}n_{k}$, where all others are zero. The latent relative mass trick of James et al. [18] can be used to include *U* as a latent variable in the likelihood for the NGG and the PYD. Setting U=1 and dividing by *N* in this case restores the posterior to the original Poisson version. The likelihood for a PYD also includes *M* (via Lemma 1). To express this, the remainder terms for both the base distribution and the CRM need to be represented.
$$\begin{array}{ccc} \lambda_{R} & = & \sum_{k=K+1}^{\infty }\lambda_{k}=\Lambda -\sum_{k=1}^{K}\lambda_{k}\\ \mu_{R} & = & \sum_{k=K+1}^{\infty }\mu_{k}\;. \end{array}$$

The joint posterior for the NGG is now
(19)$$\begin{array}{c} \mathrm{Pr}(\left\{\lambda_{k},n_{k},x_{k}\;:\;k=1,\ldots ,K\right\},U,\lambda_{R}|\mathrm{GGP},M,\alpha ,\beta ,N,\mu \left(\cdot \right))\\ =\frac{1}{N^{1_{U\neq 1}}\Gamma \left(N\right)}e^{-U\Lambda }U^{N-1}\;\;\mathrm{Pr}(\lambda_{R}|\mathrm{Twe}\left(\alpha ,{\left(M\mu_{R}\right)}^{1/\alpha },1\right))\\ \;\;\;\prod_{k=1}^{K}\lambda_{k}^{n_{k}}\mathrm{Pr}(\lambda_{k}|\mathrm{Twe}\left(\alpha ,{\left(M\mu_{k}\right)}^{1/\alpha },1\right))\\ =\frac{1}{\Gamma (N)}e^{-M\left({\left(1+U\right)}^{\alpha }-1\right)}U^{N-1}\;\;\mathrm{Pr}(\lambda_{R}|\mathrm{Twe}\left(\alpha ,{\left(M\mu_{R}\right)}^{1/\alpha },1+U\right))\\ \;\;\;\prod_{k=1}^{K}\lambda_{k}^{n_{k}}\mathrm{Pr}(\lambda_{k}|\mathrm{Twe}\left(\alpha ,{\left(M\mu_{k}\right)}^{1/\alpha },1+U\right))\;, \end{array}$$
where the second line is obtained by applying the exponential tilting formula. Note, Lemma 2 means element-wise application of a distribution to the parameter vector $\vec{\mu }$ inside μ(·). Forms for the PYD are obtained by adding the prior for *M*. For the normalised stable process, denoted NSP, one obtains
(20)$$\begin{array}{c} \mathrm{Pr}(\left\{\lambda_{k},n_{k},x_{k}\;:\;k=1,\ldots ,K\right\},U,\lambda_{R}|\mathrm{NSP},M,\alpha ,N,\mu \left(\cdot \right))\\ =\frac{1}{\Gamma (N)}e^{-U\Lambda }U^{N-1}\;\;\mathrm{Pr}(\lambda_{R}|\mathrm{pstable}\left(\alpha ,{\left(M\mu_{R}\right)}^{1/\alpha }\right))\\ \;\;\;\prod_{k=1}^{K}\lambda_{k}^{n_{k}}\mathrm{Pr}(\lambda_{k}|\mathrm{pstable}\left(\alpha ,{\left(M\mu_{k}\right)}^{1/\alpha }\right))\\ =\frac{1}{\Gamma (N)}e^{-MU^{\alpha }}U^{N-1}\;\;\mathrm{Pr}(\lambda_{R}|\mathrm{Twe}\left(\alpha ,{\left(M\mu_{R}\right)}^{1/\alpha },U\right))\\ \;\;\;\prod_{k=1}^{K}\lambda_{k}^{n_{k}}\mathrm{Pr}(\lambda_{k}|\mathrm{Twe}\left(\alpha ,{\left(M\mu_{k}\right)}^{1/\alpha },U\right))\;. \end{array}$$

From this, one can derive an integral formula for the PYD. Details are in Appendix C, and the result is original.

**Lemma** **6.**(Integral formula for the PY distribution) *Let $\vec{\mu }$ be a K-dimensional non-zero probability vector. Then, consider $\vec{\theta }∼PYD\left(\alpha ,\beta ,\vec{\mu }\right)$ for α>0 and β≥0. To express the probability of $\vec{\theta }$, introduce corresponding latent variables $\vec{\nu }=\left(\nu_{1},\ldots ,\nu_{K}\right)\in {\left[0,\pi \right]}^{K}$:*
(21)$$\mathrm{Pr}(\vec{\theta }|PYD,\alpha ,\beta ,\vec{\mu })=\int_{{\left[0,\pi \right]}^{K}}\frac{\alpha^{K-1}\Gamma \left(1+\beta \right)}{{\left(1-\alpha \right)}^{K-1}\pi^{K}\Gamma \left(1+\beta /\alpha \right)}\frac{\Gamma \left(K+\beta \left(1-\alpha \right)/\alpha \right)\prod_{k=1}^{K}a_{\alpha }\left(\nu_{k}\right){\left(\frac{\mu_{k}}{\theta_{k}}\right)}^{1/(1-\alpha )}}{{\left(\sum_{k=1}^{K}a_{\alpha }\left(\nu_{k}\right){\left(\frac{\mu_{k}}{\theta_{k}}\right)}^{1/(1-\alpha )}\theta_{k}\right)}^{K+\beta (1-\alpha )/\alpha }}d\;\vec{\nu }\;.$$

This can be readily evaluated using numerical integration for small *K*. Plots of the marginal for $\theta_{1}$ for different parameter settings are given in Figure 1 and Figure 2.

Due to the aggregation property of the PYD, these are representative marginals of the distribution for all dimensions. One can see the distributions becoming increasingly skewed as α increases.

## 6. Networks of Processes

The next natural question to consider is how the above results apply to networks of processes. Several general schemes have been developed for inference in more general networks [3,6,11,19,39,40]. General networks for HPYPs have been demonstrated to scale [4,6], in contrast to earlier Gibbs schemes [3,40], and arguably the HGP has advantages over the HDP [11]. This section is a review of related material with regards to hierarchical processes.

### 6.1. Identifiability

One important question is the issue of statistical identifiability, and an underlying issue here is whether the parametric structure admits a unique representation [41]. In our case, some simple classes of non-uniqueness are easily identified and avoided. For instance, in Poisson matrix factorisation, if the matrix entry $x_{i,j}∼Poisson\left(\sum_{k=1}^{K}\theta_{i,k}\phi_{k,j}\right)$, then one can insist that the scale of one of the matrices $\vec{\Theta }$ or $\vec{\Phi }$ (comprising the entries $\theta_{i,k}$ and $\phi_{k,j}$ respectively) needs to be anchored somehow so that the scale of the Poisson parameter is uniquely determined by just the other one. So, the rows of one of the matrices should normalise.

### 6.2. Equivalences

Another issue is that in some cases, networks can be transformed from one case to another. For instance, Zhou and Carin [11] ([Section VB]) show that a Poisson gamma-gamma process construction is equivalent to a HDP construction with an independent Poisson-gamma on the total. Given that there are significant differences between the corresponding algorithms in this case, and there are many more in the literature, what other equivalences are there?

Normalising processes are conducted to convert a CRM into an NRMI and in some cases, independence between the parts yields an equivalence between the CRM form and the NRMI form augmented with a total. This has major implications to networks of such processes, presented in the following subsection, so the results are summarised here.

The first results are on discrete processes and are well-known, some for instance reproduced by Zhou and Carin [11].

**Lemma** **7.**(Equivalent processes) *Let $\vec{\mu }$ be a probability vector (possibly infinite), and M be a constant positive background rate. Let $X=\sum_{i=1}^{\infty }x_{i}$, the sum of entries of the non-negative integer vector $\vec{x}$. The following equivalences between (A) and (B) hold:*

*Conditioning the PPP,*

$$(A)\;\vec{x}∼PP\left(M\vec{\mu }\right)\;\;\;(B)\;X∼Poisson\left(M\right)and\vec{x}∼MP\left(X,\vec{\mu }\right)\;.$$


*NBP as a Poisson-gamma mixture,*

$$(A)\;\vec{x}∼NBP\left(M,\rho ,\vec{\mu }\right)\;\;\;(B)\;\vec{x}∼PP\left(GP\left(M,\frac{1-\rho }{\rho },\vec{\mu }\right)\right)\;.$$


*DCMP, given $X\in N^{+}$, as a multinomial-Dirichlet mixture,*

$$(A)\;\vec{x}∼DCMP\left(X,M\vec{\mu }\right)\;\;\;(B)\;\vec{x}∼MP\left(X,DP\left(M\vec{\mu }\right)\right)\;.$$


*Conditioning the NBP,*

$$(A)\;\vec{x}∼NBP\left(M,\rho ,\vec{\mu }\right)\;\;\;(B)\;X∼NB\left(M,\rho \right)\;and\;\vec{x}∼DCMP\left(X,M\vec{\mu }\right)\;.$$




The conditioned versions of the PPP and NBP are used to decompose a likelihood into a total count and the vector of counts for atoms, given the total. Notice, while the conditioned version of the PPP yields a likelihood where the normalised measure ($\vec{\mu }$) and its total (*M*) are independent, the same does not hold for the conditioned NBP.

### 6.3. Normalisation and Independence

On non-discrete processes, some independences apply.

**Lemma** **8.**(Normalised processes and independence) *Let $\Lambda =\sum_{i=1}^{\infty }\lambda_{i}$, the sum of entries of the infinite non-negative real vector $\vec{\lambda }$. The following two pairs (A) and (B) are equivalent:*

*For the gamma process:*
 *(A)* 
*$\vec{\lambda }∼GP\left(M,\beta \right)$;*
 *(B)* 
*Λ∼ga(M,β) and $\vec{\lambda }/\Lambda ∼GEM\left(0,M\right)$, where *Λ* and $\vec{\lambda }/\Lambda$ are independent.*

*For the generalised gamma process where 0<α<1, marginalising M*
 *(A)* 
*$\vec{\lambda }∼GGP\left(M,\alpha ,\beta \right)$ where $M∼ga(\delta /\alpha ,\beta^{\alpha })$;*
 *(B)* 
*Λ∼ga(δ,β) and $\vec{\lambda }/\Lambda ∼GEM\left(\alpha ,\delta \right)$, where *Λ* and $\vec{\lambda }/\Lambda$ are independent.*


*Moreover, the gamma process is the only case of such independence possible for pure NRMIs (this excludes the second case as it is marginalised).*


Independence in the PYP case (represented as GEM(α,δ) in the lemma) is shown by Pitman and Yor [28] ([Proposition 21]).

That the gamma process is the only independence case for CRMs and their NRMIs is a result by Perman et al. [27] ([Corollary 2.3]). This is equivalent to the neutrality of the Dirichlet distribution, again the only distribution on probability vectors exhibiting neutrality. Neutrality and independence in this case can be shown to be equivalent properties. Independence in both these cases is also a consequence of the fact that so-called sized-biased sampling for the cases is independent of the total [27,29]. Independence properties such as in Lemma 8 do not hold generally, as indeed sized-biased sampling is not generally independent of the total.

**Lemma** **9.**(Normalisation of other process) *Let $\Lambda =\sum_{i=1}^{\infty }\lambda_{i}$, the sum of entries of the infinite vector $\vec{\lambda }$.*

*For the generalised gamma process, if $\vec{\lambda }∼GGP\left(M,\alpha ,\beta \right)$ then $\Lambda ∼Twe(\alpha ,M^{1/\alpha },\beta )$ and $\vec{\lambda }/\Lambda ∼NGG\left(\alpha ,M\right)$.*

*For the stable process, if $\vec{\lambda }∼staP\left(M,\alpha \right)$ then $\Lambda ∼pstable(\alpha ,M^{1/\alpha })$ and $\vec{\lambda }/\Lambda ∼PYP\left(\alpha ,0\right)$,*


*Λ*
*and $\vec{\lambda }/\Lambda$ are not independent in either case.*


### 6.4. Modelling LDA Using HDP

Consider models for the HDP variant of LDA [3], called HDP-LDA, which has been the subject of extensive research. There is a wide variation in the literature of how these are to be represented by graphical model and for statistical inference. Figure 3 shows two equivalent models for HDP-LDA. Figure 3a gives the original model as formulated by Teh et al. [3], and Figure 3b shows the modification used here. Authors sometimes use a more complicated formulation in terms of the underlying stick-breaking model.

In this problem, there are *D* documents and $N_{d}$ words in each document for d=1,…,D, where the words $w_{d,n}$ are modelled with an admixture. The probabilistic specification for the corresponding models are given in Figure 4.

Figure 4a shows the probabilistic specification with full base distributions. While this follows the theory directly, it is a fairly large departure from the original representation of LDA. The reformulation in Figure 4b is a direct analogue of the original representation of LDA with two modifications essential for the treatment of a HDP, discussed below as the root node and the non-root node.

The root node of the DP hierarchy is represented as a GEM, which generates the infinite vector. In practice, this can be represented using size-biased sampling [27] formulations, and in the simplest and popular cases this corresponds to stick-breaking methods [42]. In implementation, however, there is no need for this as posterior formulations for the processes are well understood and require no implicit ordering constraints as in stick-breaking.

Non-root nodes down the hierarchy are represented using their underlying infinitely divisible non-negative distribution, in this case the Dirichlet. Note, however, this extends the standard definition of a Dirichlet as the input parameter is an infinite dimensional vector. In implementation, this is no impediment as only a finite amount of data is ever dealt with, although it does require modelling the current number of non-empty dimensions. This can be readily handled using standard parametric techniques [6] or by using truncation [4].

Note Figure 4a also uses a nested construction [43] with the expression $\mathrm{DP}(c_{\alpha },\mathrm{Dirichlet}\left(c_{\beta }\vec{\beta }\right))$. Here a distribution, in this case a Dirichlet, but it could also be a GP, a DP or any other process, is used as the base distribution. This nesting construction is exactly what is needed to model matrix and tensor factorisation using hierarchical processes.

The nested, hierarchical equivalent to Figure 5b is as follows:$$\begin{array}{ccc} \vec{\beta } & \;\;∼\;\; & GEM(d_{\beta },c_{\beta })\\ G_{0} & \;\;∼\;\; & GP\left(c_{\alpha },1,\mathrm{PYD}\left(d_{\phi },c_{\phi },\vec{\beta }\right)\right)\\ G_{d} & \;\;∼\;\; & GP(c_{\theta },s_{\theta },G_{0})\\ \phi_{d,n} & ∼ & G_{d}\\ {\vec{n}}_{d} & \;\;∼\;\; & Poisson\left({\vec{\phi }}_{d}\right) \end{array}$$

The background word probabilities $\vec{\beta }$ are generated, then used as the base distribution for a PYD which then creates variants ${\vec{\phi }}_{k}$ as each atom of the gamma process $G_{0}$. The mixture weights of $G_{0}$ correspond to $\vec{\alpha }$ from Figure 5b. Variants of this, $G_{d}$, are then created which modify the mixture weights $\vec{\alpha }$ but leave the atoms constant. So, $G_{d}$ is a weighted sum of the original ${\vec{\phi }}_{k}$, as is the case in Figure 5b. This is very elegant, but Figure 5b better exposes the detail needed for implementation.

### 6.5. Example Equivalences with Non-Parametric LDA

Consider extending HDP-LDA to include a Pitman–Yor distribution on the word side. This model, termed NP-LDA Buntine and Mishra [4], has been demonstrated using a truncated approximation. To bring out equivalences, the multinomial form of the topic model is given, and both are defined in Figure 5.

The gamma scale parameter on $\alpha_{0}$ is one as it has an equivalent affect to $c_{\theta }$. So, it needs to be made a constant for identifiability. The equivalence is obtained by noting, from Lemmas 7 and 8, and many such results exist for the finite case, for instance by [44]. One can introduce a total rate for documents, $\Theta_{d}$, and model the count, $N_{d}$, entirely independently:$$\begin{array}{ccc} \alpha_{0} & ∼ & gamma(c_{\alpha },1)\\ \Theta_{d} & ∼ & gamma(c_{\theta }\alpha_{0},s_{\theta })\\ N_{d} & ∼ & Poisson(\Theta_{d})\;. \end{array}$$
If the concentration parameters are estimated during learning, which is the common case, and recommended for topic models, then equivalence does not hold.

Experimental evidence [4] shows the following:The topic side, ${\vec{\theta }}_{d}$, is best not modelled using PYPs because experiments indicate that this gives no performance improvement. The non-Zipfian DPs work best, probably because of the smaller dimensions for number of topics.Modelling the word side, ${\vec{\phi }}_{k}$, using PYPs systematically outperforms HDP-LDA by a moderate margin in perplexity and yields more explainable topics because the overall “background” words are separately modelled using $\vec{\beta }$.

Several model equivalences hold with regard to these kinds of models.

The asymmetric-symmetric version of LDA [45] is a truncated version, not well understood in the community.The asymmetric-asymmetric version of LDA, evaluated by Wallach et al. [45], is a truncated version of the model in Figure 5a.Hierarchical Poisson factorisation [46] (HPF) is a non-parametric formulation of Poisson-gamma matrix factorisation using stick-breaking, and thus is equivalent to HDP-LDA above (when augmented with a gamma model of the total counts).Robust (negative binomial) Poisson factorisation by Zhou et al. [47] is related (ignoring some issue with hyperparameters) to bursty topic models by Doyle and Elkan [48], which has a non-parametric extension in Buntine and Mishra [4].

## 7. Conclusions

Discrete base distributions make CRMs behave like vectors of infinitely divisible distributions, where application is element-wise without the non-parametrics. So, the gamma process becomes an element-wise gamma distribution, and the generalised gamma process becomes an element-wise Tweedie distribution. This was presented in Lemma 2, Lemma 4 and Corollary 1 and accompanying tables. Similarly, discrete base distributions make NRMIs and related processes behave as normalised versions of the above, sharing some properties of the DP such as renormalisation. So, the HPYP becomes the PY distribution, whose form was developed in Section 5.

If closed forms for analysis of the infinitely divisible distributions do not exist, the generalised versions of Chinese restaurant process (CRP) sampling, given in Equation (Equation 17), can be used instead, including versions of the more recent, efficient collapsed samplers for CRPs [6,39], given in Equation (Equation 18). Similar formulations also appear in [14,15]. Note many of these quantities, for instance in Table 1, can be derived from the Laplace exponent of the CRM, so a convenient form of the distribution is not needed. The CRPs come about when unfolding the recursion that relates the cumulants of a distribution to the moments of the distribution, a simple result in basic statistics. In this way, known CRPs for the gamma process follow a general scheme that also applies for the generalised gamma process, the generalised beta process and others.

While most of these results follow fairly simply from general results in the non-parametric Bayesian community, some have not yet seen use in the Bayesian machine learning community.

As a specific example of hierarchical distributions, it was also shown in Section 5 that the NGG and PY distributions, for the case where discount α>0 and concentration β>0, are behaving like normalised Tweedie variables, and for the case where concentration β=0 like normalised positive stable variables. Moments of the Tweedie distribution show how the standard hierarchical likelihood for the HPYP used to date [5,37] can be directly derived from this framework without considering non-parametric theory. A novel integral expression for the PY distribution for discount α>0 and concentration β≥0 was also developed in Equation (Equation 21). This answers the question, “what is a hierarchical PYP”?

There are a rich number of variations of matrix factorisation and topic models that exist, for instance, see ([11] Table 1) with seven different versions of negative binomial matrix factorisation, and the software used in Buntine and Mishra [4] has seven different non-parametric versions of LDA. This is ignoring the extensions of the model where the problem is changed significantly: document segmentation [7], hierarchical topics [8], supervised topic models, etc., and these extensions no doubt have their own rich variety of versions and equivalences. Moreover, some of the known equivalences between processes, when applied in the hierarchical case, yield relationships between models and algorithms in the machine learning community that deserve further investigation, discussed in Section 6. This is confounded by the fact that variants are evaluated using significantly different methodologies; compare, for instance, topic modelling evaluation with recommender systems evaluation. It is an open question as to what other significant equivalences exist in the literature, and the implications this has to the algorithms one can use.

## Figures and Tables

**Figure 1 entropy-24-01703-f001:**
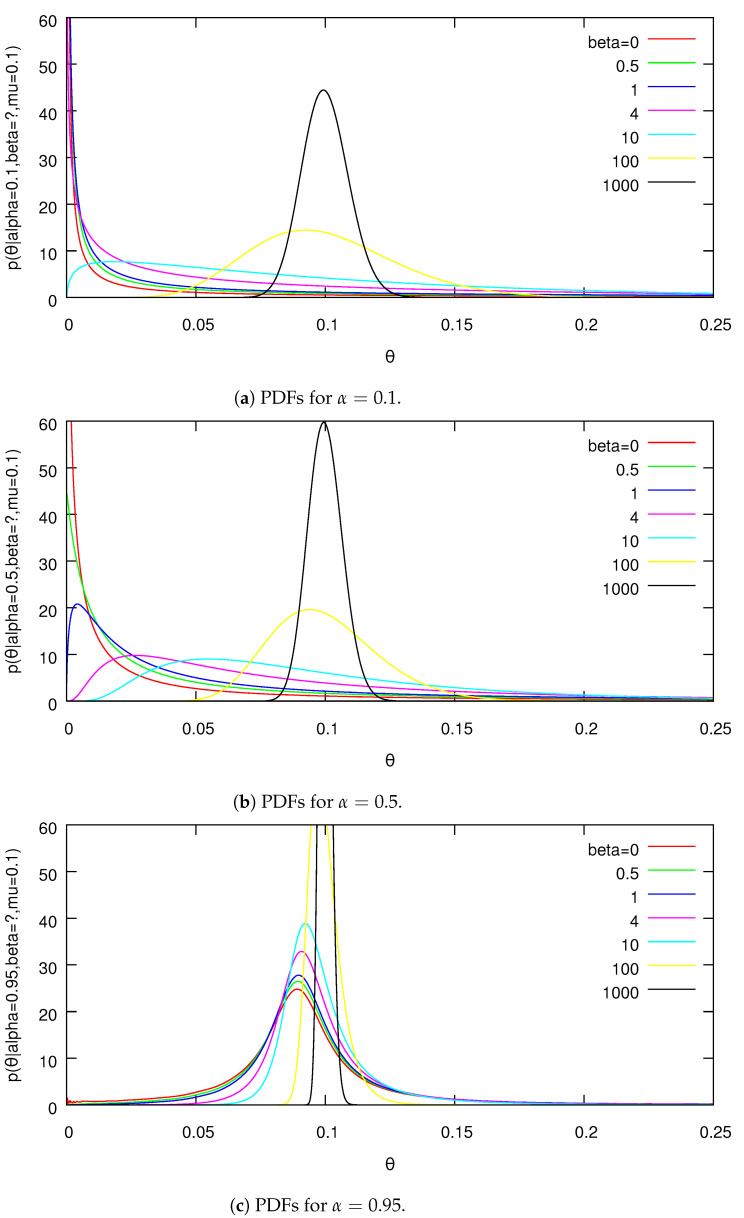
PDFs from Lemma 6 for location $\mu_{1}=0.1$ and fixed α.

**Figure 2 entropy-24-01703-f002:**
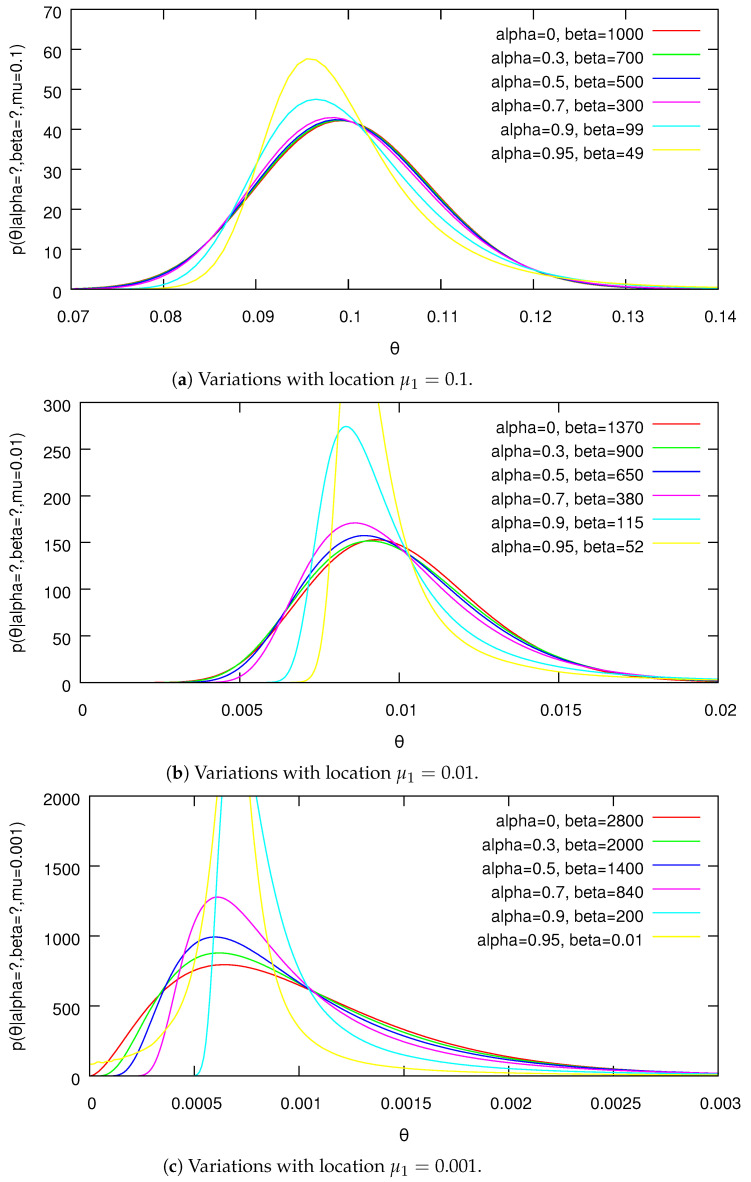
PDFs from Lemma 6 for variations with identical location and variance.

**Figure 3 entropy-24-01703-f003:**
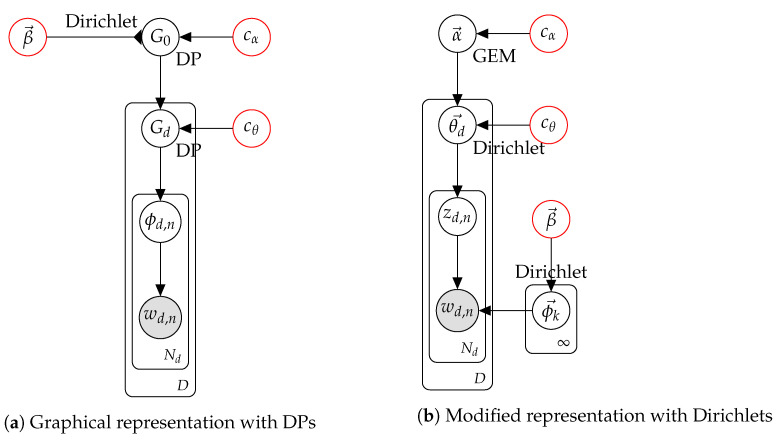
Equivalent versions of HDP-LDA. In (**a**), the arc from $\vec{\beta }$ has a modified head to indicate that $\mathrm{Dirichlet}(\vec{\beta })$ is used in a nested manner.

**Figure 4 entropy-24-01703-f004:**
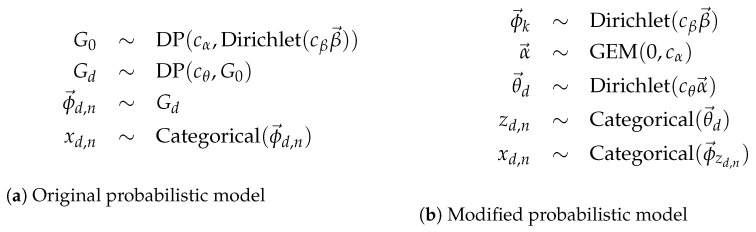
Equivalent versions of HDP-LDA. Concentration parameters $c_{X}$ treated as constants or estimated. Indices d=1,…,D, $n=1,\ldots ,N_{d}$ and k=1,…,∞. The $\vec{\phi }$ are indexed differently in the two versions. The $\vec{\alpha }$ and ${\vec{\theta }}_{d}$ are infinite probability vectors in the CRM representation of $G_{0}$ and $G_{d}$, respectively.

**Figure 5 entropy-24-01703-f005:**
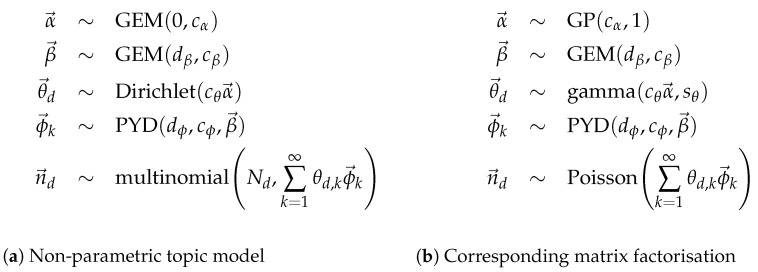
NP-LDA and its matrix factorisation counterpart. Concentration parameters $c_{X}$ are constants or estimated, as are discounts $d_{X}$. Indices d=1,…,D and k=1,…,∞. Vector-wise versions of the gamma and Poisson represent the gamma process and Poisson process, respectively.

**Table 1 entropy-24-01703-t001:** General processes. Marginal is the corresponding infinitely divisible distribution for the total rate developed, for instance, using Theorem 1.

Name	Domain	Parameters	Rate (Lévy Measure)	Marginal
beP(M,α,β)	0<λ<1	0≤α<1, β>0	$M\frac{\lambda^{-\alpha -1}}{\Gamma (1-\alpha )}{\left(1-\lambda \right)}^{\alpha +\beta -1}$	for α = 0, β = 1: Dickman(M)
GP(M,β)	λ>0	β>0	$M\lambda^{-1}e^{-\lambda \beta }$	gamma(M,β)
GGP(M,α,β)	λ>0	0<α<1, β>0	$M\frac{\alpha }{\Gamma (1-\alpha )}\lambda^{-\alpha -1}e^{-\lambda \beta }$	$Twe(\alpha ,M^{1/\alpha },\beta )$
staP(M,α)	λ>0	α>0	$\frac{M\alpha }{\Gamma (1-\alpha )}\lambda^{-\alpha -1}$	$pstable(\alpha ,M^{1/\alpha })$
PP(M)	λ=1		M	Poisson(M)
NBP(M,ρ)	$\lambda \in N^{+}$	0<ρ<1	$M\frac{-\rho^{\lambda }}{\lambda \log(1-\rho )}$	NB(M,ρ)

**Table 2 entropy-24-01703-t002:** Key formula for posterior analysis of CRMs, $\Psi_{J}=\int \mathrm{Pr}({\vec{n}}_{1:J}\neq \vec{0}|\lambda )\rho \left(d\lambda \right)$, and the distribution on the remainder $T_{R}=\sum_{i=I+1}^{\infty }\lambda_{i}$.

Name	$\Psi_{J}$	Remainder $T_{R}$
beP(M,α,β)-BP	$M\sum_{j=0}^{J-1}\frac{\Gamma (\alpha +\beta +j)}{\Gamma (1+\beta +j)}$	$\mu_{R}∼\mathrm{beP}\left(M,\alpha ,J+\beta \right)$
GP(M,β)-PP	$M\left(\log(J+\beta )-\log\beta \right)$	gamma(M,J+β)
GGP(M,α,β)-PP	$M\left({\left(J+\beta \right)}^{\alpha }-\beta^{\alpha }\right)$	$Twe(\alpha ,M^{1/\alpha },J+\beta )$
GP(M,β)-NBP(ρ)	$M\left(\log(J\log\frac{1}{1-\rho }+\beta )-\log\beta \right)$	$gamma(M,J\log\frac{1}{1-\rho }+\beta )$
GGP(M,α,β)-NBP(ρ)	$M\left({\left(J\log\frac{1}{1-\rho }+\beta \right)}^{\alpha }-\beta^{\alpha }\right)$	$Twe(\alpha ,M^{1/\alpha },J\log\frac{1}{1-\rho }+\beta )$
staP(M,α)-PP	$MJ^{\alpha }$	$Twe(\alpha ,M^{1/\alpha },J+\beta )$
staP(M,α)-NBP(ρ)	$M{\left(J\log\frac{1}{1-\rho }\right)}^{\alpha }$	$Twe(\alpha ,M^{1/\alpha },J\log\frac{1}{1-\rho })$

**Table 3 entropy-24-01703-t003:** Properties of processes.

Name	$\kappa_{n}$	ψ(t)	$T_{K}^{n}$
beP(M,α,β)	$M\frac{\Gamma (n-\alpha )}{\Gamma (1-\alpha )}\frac{\Gamma (\alpha +\beta )}{\Gamma (n+\beta )}$	$M\frac{\Gamma (\alpha +\beta )}{\Gamma (\beta )\alpha }(\left.{}_{1}F_{1}\right.\;\left(1-\alpha ,\beta ,t\right)-1$	use Equation (Equation 8)
(for β>1−α)		$+\frac{1}{\beta }\left.{}_{1}F_{1}\right.\;\left(1-\alpha ,\beta +1,t\right))$	
GP(M,β)	$M\frac{\Gamma (n)}{\beta^{n}}$	Mlog(1+t/β)	$\frac{M^{K}}{\beta^{n}}S_{K}^{n}$
GGP(M,α,β)	$M\frac{\Gamma (n-\alpha )}{\Gamma (1-\alpha )}\frac{\alpha }{\beta^{n-\alpha }}$	$M\left({\left(\beta +t\right)}^{\alpha }-\beta^{\alpha }\right)$	$\frac{{\left(M\alpha \beta^{\alpha }\right)}^{K}}{\beta^{n}}S_{K,\alpha }^{n}$
staP(M,α)	NA	$Mt^{\alpha }$	NA

## Data Availability

Not applicable.

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
