# Peer review of "Understanding Hierarchical Processes"

_entropy, 2022, doi:10.3390/e24121703_

Round 1
Reviewer 1 Report
The manuscript seeks to give a more accessible account of random processes that are used in "non-parametric" Bayesian hierarchical processes. Here, as noted by the author, hierarchical process essentially means taking a discrete random measure and composing it with another discrete random measure.
Generally, in the literature and in the present paper, these are CRM or normalizations of such processes, depending on the application. The author wishes to give a simpler account of these processes.
I think overall the manuscript is fine. I however do have some comments and suggestions
1. Indeed the notion of hierarchical processes, within this context, is related to the idea of a random process whose "base measure" is also a discrete random process. While some references are given in the introduction where such processes are used it would be good to give a more detailed explanation/example earlier as to what discrete base measures actually do. Why are they used? What is the modelling aspects of using them?
Otherwise it seems the reader would have to get that motivation from reading other papers.
My remaining comments are directly linked to the text.
Lines 66-69. The notion of creating random probability measures by Normalizing subordinators (CRM) goes back the the paper of Kingman (1975)--Kingman, J.F.C. (1975), Random Discrete Distributions. Journal of the Royal Statistical Society: Series B (Methodological), 37: 1-15. (The name Poisson-Kingman is related to this) The reference [15] revitalized this idea, Simple to say, see also Kingman (1975).
Lines 112-113 the use of the GEM sequence refers to the size biased order of the masses of a Dirichlet process. The representation in equation (1) typically is based on ranked order of the points. So the statement in lines 112-113 needs to be more precise (See also my comment below about lines 280-284_
Line 142-- do you mean "generalized gamma" instead of just "gamma"?
Line 148 Theorem 1--- it should be "\sigma_t "has" an infinitey divisible...
In Definition 2 p. 5. The use of \lamba_k has a different meaning than its use in equation (1) . I would suggest to use different notation
Lines 205-206, I think you may need to be clear and say "in the relevant non-hierarchical settings" or something similar to distinguish from results for hierarchical versions. This is more clearly stated in lines 37-40.
Lines 280-284 The stick-breaking construction, again refers to the size-biased representation (see Perman, Pitman and Yor). Furthermore there are only a few explicit cases.
page 10 Table 3. I do not believe the expression for \psi(t) in the beta process case is correct. That is , it is since the levy density is has infinite total mass, you cannot write psi(t) as the "confluent hypergeometric function-1." Please check this.
Lines 452-453 Independence is due to Proposition 21 of Pitman and Yor (1997)
Lastly you should consider if the following references are worth citing,
Argiento, R., Cremaschi, A. and Vannucci, M. (2020) “Hierarchical Normalized Completely Random Measures to Cluster Grouped Data”, Journal of the American Statistical Association, volume 115 issue 529, pp 318-333.
Camerlenghi, F., Lijoi, A., Orbanz, P., and Prunster, I. (2018). Distribution theory for hierarchical processes. The Annals of Statistics, 47, 67-92.
Author Response
Indeed the notion of hierarchical processes, within this context, is related to the idea of a random process whose "base measure" is also a discrete random process. While some references are given in the introduction where such processes are used it would be good to give a more detailed explanation/example earlier as to what discrete base measures actually do. Why are they used? What is the modelling aspects of using them?
Excellent point. I added a new second paragraph.
Lines 66-69. The notion of creating random probability measures by Normalizing subordinators (CRM) goes back the the paper of Kingman (1975)--Kingman, J.F.C. (1975), Random Discrete Distributions. Journal of the Royal Statistical Society: Series B (Methodological), 37: 1-15. (The name Poisson-Kingman is related to this) The reference [15] revitalized this idea, Simple to say, see also Kingman (1975).
Done.
Lines 112-113 the use of the GEM sequence refers to the size biased order of the masses of a Dirichlet process. The representation in equation (1) typically is based on ranked order of the points. So the statement in lines 112-113 needs to be more precise
Corrected.
Line 142-- do you mean "generalized gamma" instead of just "gamma"?
Correct.
Line 148 Theorem 1--- it should be "\sigma_t "has" an infinitey divisible…
Correct.
In Definition 2 p. 5. The use of \lamba_k has a different meaning than its use in equation (1) . I would suggest to use different notation
Done. Good catch.
Lines 205-206, I think you may need to be clear and say "in the relevant non-hierarchical settings" or something similar to distinguish from results for hierarchical versions. This is more clearly stated in lines 37-40.
Done.
Lines 280-284 The stick-breaking construction, again refers to the size-biased representation (see Perman, Pitman and Yor). Furthermore there are only a few explicit cases.
Noted.
page 10 Table 3. I do not believe the expression for \psi(t) in the beta process case is correct. That is , it is since the levy density is has infinite total mass, you cannot write psi(t) as the "confluent hypergeometric function-1."
The Levy density does have infinite mass around zero, but it satisfies the usual conditions given in Theorem 1, and the Laplace exponent exists. But as you point out, this hypergeometric function is usually not defined for negative first argument, that is F(-a,b,t) is not usually allowed. I corrected for this by doing integration by parts and re-expressing in terms of legal values F(1-a,b,t) and F(1-a,b+1,t).
Lines 452-453 Independence is due to Proposition 21 of Pitman and Yor (1997)
Fixed.
References "Hierarchical Normalized Completely Random Measures to Cluster Grouped Data\" by Ariento et al. 2020 and "Distribution theory for hierarchical processes" by Camerlenghi et al. 2019.
These are indeed highly relevant and I was not aware of them. My theory was developed in 2017 on a sabbatical before these were available, and I distributed it privately but didn't publish. To accommodate these important papers I have modified the last two paragraphs in the introduction and several paragraphs in the conclusion. I have also cited the work were relevant theorems/lemmas are given, for instance Lemma 4, Lemma 5 and Corollary 1.
Reviewer 2 Report
Overall, the manuscript is well organised, clear, and well written. It has a directed focus in line with a literature review of hierarchical stochastic processes and their applications.
The manuscript outlines in a clear and well-structured way what are the essential properties and features of these (hierarchical) processes, as well as useful representations that can aid in computation and/or simulation.
In my view, the contribution of the manuscript is as a review on the topic (I am not aware of a similar one on the same topic), but it does contain some derivations (Lemmas 4 and 6, and Corollary 1.) It should perhaps be highlighted more clearly if these are original contributions or merely routine computations that follow from a more general framework.
In Definition 5, "completely neutrality" does not seem very grammatical; contrast with Definition 2 of James [30].
Author Response
In Definition 5, "completely neutrality" does not seem very grammatical; contrast with Definition 2 of James [30].
Thank you for the correction. This has been fixed.
it does contain some derivations (Lemmas 4 and 6, and Corollary 1.) It should perhaps be highlighted more clearly if these are original contributions or merely routine computations that follow from a more general framework.
This is a good suggestion. Appropriate comments have been added directly before these lemmas and the corollary.